# Time to First Fix Robustness of Global Navigation Satellite Systems: Comparison Study

**DOI:** 10.3390/s25051599

**Published:** 2025-03-05

**Authors:** Carlos Hernando-Ramiro, Óscar Gamallo-Palomares, Javier Junquera-Sánchez, José Antonio Gómez-Sánchez

**Affiliations:** Spanish National Institute for Aerospace Technology (INTA), Carretera de Ajalvir, km 4, 28850 Torrejón de Ardoz, Spain

**Keywords:** cold start, global navigation satellite system (GNSS), low-cost receiver, mass-market applications, signal attenuation, single frequency, time to first fix

## Abstract

The time to first fix (TTFF) measures the time elapsed by a global navigation satellite system (GNSS) receiver from switch-on to provision of a navigation solution. This parameter is crucial for applications where a position, within an acceptable error, is needed as soon as possible after turning the device on. The quality of the TTFF depends mainly on the receiver, the environment, and the GNSS satellites employed. Although all four available GNSSs (BeiDou, Galileo, GLONASS, and GPS) are complementary, their constellations and signals differ, providing different TTFF performances. This becomes even more relevant in hostile environments, where the TTFF degrades from nominal results. In this work, the robustness of the signals of the four GNSSs against different levels of harshness and its influence on the TTFF performance are evaluated in a comparative way. For this purpose, a typical scenario for mass-market GNSS applications, involving cold-start conditions, single-frequency signals, and a low-cost receiver, is considered. The results indicate that GPS provides the most robust TTFF, followed by GLONASS (although at the expense of positioning accuracy), BeiDou, and Galileo, in that order.

## 1. Introduction

A global navigation satellite system (GNSS) provides a constellation of satellites that transmit location and timing signals to any part of the globe, on land, sea, or air, allowing the receiving equipment to determine its coordinates, altitude, velocity, and current time with great accuracy, 24 h a day and in all weather conditions [1]. The availability of this position, velocity, and time (PVT) solution enables the development of navigation, tracking, transport, synchronisation, emergency, geomatics, and agricultural applications, among others [2]. Nowadays, there are four GNSSs available worldwide, each developed and governed by a different country or union of countries. The first GNSS that became fully operational, in 1993, was the Global Positioning System (GPS), originally Navstar GPS, which is owned by the United States [3]. Afterwards, in 1995, Russia completed its GNSS, which is called GLONASS [4]. Later, in the 21st century, both China and the European Union each deployed their own GNSSs, which are named BeiDou [5] and Galileo [6], respectively.

From the user perspective, all four GNSSs are complementary, meaning that signals from any constellation can be used together to compute a PVT solution [7,8]. For this purpose, all of them provide signals with the same type of information, but the technology for their generation and transmission varies from one to another [9]. This implies that, for a receiver, it may be easier to acquire and track signals to calculate its first position when using a GNSS in comparison to the others, mainly under adverse conditions. This aspect is especially important for the low-cost GNSS chipsets of mass-market devices, which usually undergo economic, size, or power constraints. These receivers may not be constantly powered and, for the sake of simplicity and legacy, tend to work with the standard frequency band and do not implement countermeasures against signal deterioration. This is the case of market segments such as smart agriculture, wildlife tracking, health and lifestyle, search and rescue, emergency services, smart cities, intelligent logistics, and environmental monitoring [10]. For this reason, the aim of this study is to assess the robustness of the time to first fix of the different GNSSs, analysing the performance of a low-cost receiver in providing a useful initial PVT against different levels of harshness.

The time to first fix (TTFF) is the time required for a GNSS receiver to start up, acquire satellite signals, obtain navigation data, and calculate its current position [11]. The TTFF is a parameter of high interest for those applications where having a first position fix within a few seconds is more important than waiting for a more accurate solution [12]. In this sense, for mass-market devices, a low TTFF offers significant advantages for the aforementioned segments as follows:Improved power efficiency: GNSS modules can switch off sooner after deriving a fix, reducing consumption and extending battery life. Relevant applications include wildlife tracking, intelligent logistics, and environmental monitoring [13,14,15].Enhanced user experience: Users receive positioning data with minimal delay, ensuring smoother operations. This is particularly beneficial for smart agriculture, healthcare and lifestyle, and smart cities [16,17].Faster response times: Rapid location acquisition enables quicker reactions in search and rescue operations and emergency services [18].

In this context, the TTFF robustness, i.e., the capability of obtaining a low TTFF under adverse conditions, varies among GNSSs, depending on both the number of operational satellites and the strength of their signals [19,20]. Therefore, if a GNSS offers a high quantity of healthy satellites and powerful GNSS signals, it is more likely that the receiver can rapidly detect and track at least four satellites to compute the PVT solution.

Moreover, the TTFF also depends on the starting conditions of the receiver. For instance, if the receiver already has stored navigation and time information, it can be used to reduce the TTFF, corresponding to the so-called warm and hot starts [21]. On the other hand, in this work, we consider a cold-start scenario, where there are no previous data available at the GNSS receiver and, consequently, a full search of satellites from scratch is required. This situation is more demanding but fits better with the general low-consumption needs of multiple mass-market applications, where the PVT data are not constantly derived and the GNSS chipset is only switched on when needed [22,23].

In this study, we propose assessing the robustness of the TTFF for the different GNSSs in a comparative way. For this purpose, we develop an ad hoc setup that allows us to test how easy it is for a typical low-cost receiver to obtain a useful fix with each GNSS separately. In this way, for the very same receiver and conditions, the TTFF is influenced only by the availability and quality of the GNSS signals [24,25]. As the TTFF also depends on the technological design and implementation of the receiver, the specific values obtained here should not be taken as absolute references. However, the results are valid for establishing a comparison of TTFF robustness between the different GNSSs.

## 2. Materials and Methods

In order to carry out the proposed study and achieve a fair comparison between the four GNSSs, the very same geometry of satellites and signal conditions need to be applied many times to different configurations of the GNSS receiver. For this purpose, we design and prepare the test setup depicted in Figure 1, which comprises the following elements:GNSS antenna: A full-spectrum choked-ring antenna, Leica AR20 with a dome, is employed to capture the signals of all GNSSs. The antenna is installed on the roof of a building at the campus of the Spanish National Institute for Aerospace Technology (INTA) located in Torrejón de Ardoz.GNSS recorder and playback system: Live GNSS signals are recorded and later reproduced to carry out all the test runs with a Spirent (Spirent Communications plc, Crawley, UK) GSS6450 multi-frequency record and playback system [26]. The radio-frequency spectrum that is acquired includes the main band of each GNSS, which are always available in any low-cost receiver:-BeiDou B1: 1561.098 MHz;-GPS L1 and Galileo E1: 1575.42 MHz;-GLONASS G1: 1602.018 MHz.The configuration applied in the Spirent GSS6450 for recording is as follows:-Central frequency: 1583.604 MHz;-Bandwidth: 50 MHz;-Sample resolution: 8 bits.RF attenuator: This playback system also includes an output attenuation function that allows us to simulate hypothetical degradations of the power of the signals due to external factors.GNSS receiver: It must allow the automation of all the test runs by the remote configuration and management of the following:-Cold-start execution;-Configuration of the GNSS (BeiDou, Galileo, GLONASS, or GPS) to be used;-Access to National Marine Electronics Association (NMEA) 0183 messages [27].For this aim, a representative low-cost receiver, specifically a u-blox (u-blox AG, Thalwil, Switzerland) ZED-F9P with firmware version 1.32 [28], is used [29,30,31]. This receiver tracks and acquires signals of operational satellites of all generations simultaneously. For the test, its navigation mode is configured with the stationary dynamic platform model and the automatic position fixing mode. Moreover, in order to detect as many satellites as possible, no minimum values for the elevation angle and the signal-to-noise ratio (SNR) are set in its navigation input filters. On the other hand, its navigation output filters, which control the quality of the PVT, are kept with their default values: 25 for maximum dilution of precision (DOP) and 100 m for minimum positioning accuracy.

In order to achieve representative results, the test plan covers four complete days. Specifically, a recording of 10 min is acquired every hour starting at 00:00 UTC of 4 November 2023 until 00:00 UTC of 8 November 2023. In total, 97 10-min recordings, containing the actual spectrum from 1558.604 MHz to 1608.604 MHz, are acquired. In this way, although only one location is used, many different constellation configurations of the four GNSSs are covered. The test duration and number of recordings are limited by the data storage capacity available, as the acquisition of each 10-min signal produces a file of more than 63 GB (over 6 TB in total).

Another relevant factor of the test is the hypothetical attenuation of the signal power in challenging scenarios. For this purpose, not only is the original radio-frequency spectrum used, but also several attenuated versions of it. In particular, 11 different attenuation values, from 0 dB to 20 dB in steps of 2 dB, are applied. In this way, the results show the behaviour of the GNSSs against different levels of adverse conditions.

In addition, as the behaviour of the receiver tracking satellites is not deterministic, 10 iterations are carried out with each recording and attenuation level in order to assure the reliability of the results. In total, since we use 4 GNSSs, 97 recordings, 11 attenuation levels, and 10 iterations, 42,608 test runs, that can last up to 10 min each, are carried out.

The workflow of the methodology followed in the test for each GNSS individually is provided in Figure 2, so each of its 10,652 runs is executed as follows:Selection of the recording and the attenuation level;Execution of a cold start in the receiver and playback of the signal;Acquisition of the GSV and GGA NMEA sentences until the first fix.

In this study, due to constraints in bandwidth recording, data storage, and execution time, the test is limited to the most common frequency band and a widely used low-cost receiver. For this reason, the results are only useful for comparative analysis between GNSSs, and it is not intended to provide absolute claims about them. Nonetheless, since the proposed methodology relies on general NMEA messages, it can be applied to any other recorded signals and receivers of interest.

## 3. Results

Employing the described test setup and following the presented test plan, several parameters that allow us to assess the TTFF robustness of the different GNSSs are derived. Firstly, two of the main factors that affect the TTFF, namely the number of tracked satellites and the power of the signals, are analysed. Afterwards, the TTFF itself, which is achieved with the different GNSSs, is assessed. Finally, the horizontal positioning error is also studied, as it strongly influences the usefulness of the solution. All test results are derived as the average of the 97 signals and their 10 iterations. For this purpose, data obtained from the NMEA messages (version 4.11) provided by the u-blox ZED-F9P receiver through 11 test runs, one per attenuation level, are used.

### 3.1. Tracked Satellites

Using only one GNSS, a receiver needs to track at least four satellites to derive the first fix. In this sense, the number of operational satellites, i.e., those providing useful signals, contributes to the quantity of satellites that a receiver is able to track. At the time of the acquisition of the recordings, BeiDou had 46 operational satellites in orbit, but only 34 of them were visible from the test location, 27 were following medium-Earth orbits (MEOs), 6 were following inclined geosynchronous orbits (IGSOs), and 1 had a geostationary orbit (GEO). Furthermore, Galileo had 23 operational satellites, GLONASS had 24, and GPS had 31. In addition, the robustness of the emitted signals, in terms of both power and modulation, facilitates the tracking of the satellites. In order to study this effect, the quantity of tracked satellites is obtained from the GSV (satellites in view) NMEA sentences provided by the receiver at the time of achieving its first fix.

In Figure 3, the quantity of satellites of each constellation that the receiver tracks in nominal conditions, i.e., without applying any attenuation, along the 4 days of the test, is shown. In addition, Table 1 provides statistical results of the data depicted in Figure 3. It can be observed that GPS is the system that, on average, allows the receiver to track the highest number of satellites. On the other hand, Galileo is the one with the lowest number of tracked satellites. In addition, BeiDou is the GNSS that shows the greatest variability over time in terms of tracked satellites.

The evolution of the average number of tracked satellites as a function of attenuation is depicted in Figure 4. These data show that GPS and GLONASS are affected in a similar way, losing two satellites until 20 dB of attenuation. Galileo exhibits a similar behaviour, as its tracked satellites are lessened from 5 to 3 at 18 dB. On the other hand, for BeiDou, the reduction in tracked satellites with attenuation is larger, with a steeper drop starting from 16 dB.

### 3.2. Signal Power

The power of the GNSS signals facilitates the tracking of the satellites and, consequently, the reduction in the TTFF. Therefore, the TTFF is more robust if the power of its signals is high enough to be detected and used by the receiver. Apart from the original signal power of the satellite and the constellation geometry [32], which may vary between GNSSs, several environmental factors can degrade the SNR detected at the receiver. These include atmospheric effects such as ionospheric scintillation [33], physical obstructions and multipaths caused by natural or human-made objects [34], and unintentional or intentional radio-frequency interference.

In this study, the environmental factors are identical for all GNSSs, as the same recorded spectrum is applied for each of them. Therefore, the differences in SNR between them are only due to their signals and constellation characteristics. In order to assess this, the average SNR of the different GNSSs as a function of attenuation is shown in Figure 5. These SNR values are obtained as the mean of the SNR levels of all the tracked satellites, provided by the GSV NMEA message, at the time of the first fix or at the end of the test run (10 min) if no fix is achieved. It must be noted that the set of tracked satellites may vary for different attenuation values, preventing a simple linear relationship between SNR and attenuation.

The results show that, for low attenuation levels, BeiDou exhibits the highest SNR values. However, from 8 dB onwards, GPS is the one that provides the best results. In addition, Galileo outperforms GLONASS for an attenuation higher than 2 dB and matches the behaviour of BeiDou after 10 dB. GLONASS is also able to keep an acceptable value of SNR for the highest attenuation level of 20 dB.

For further assessment, these signal power results should be analysed together with the tracked satellite data presented above. For instance, in some cases, better PVT solutions, i.e., both lower TTFF and positioning error, may be achieved by using more satellites with smaller but acceptable SNR values, rather than by employing fewer satellites with higher SNR levels.

### 3.3. Time to First Fix

The TTFF is the main target parameter of this study. For its calculation, we measure the time between the cold start and the generation of the first GGA (global positioning system fix data) NMEA sentence with a fix. As a starting point, the TTFF achieved without attenuation is presented in Figure 6. This graph illustrates the variability in the results due to the different geometries of the satellites. In this sense, Galileo is the GNSS most affected by this because, at the time of the test, its constellation was not yet completed. Please note that the statistical values of the data depicted in Figure 6 are shown in the first column of results in Table 2.

The results of the TTFF as a function of attenuation for the different GNSSs are presented in Figure 7. In addition, the corresponding statistics (mean, standard deviation, minimum, and maximum) are shown in Table 2. It should be noted that, in order to be able to derive graphs and statistics that include all cases under test, if no fix is achieved until the end of the recording, the TTFF is set to 10 min. However, the use of this upper limit for the TTFF gives rise to an artificial reduction in the standard deviation for high attenuation levels.

The outcome data reveal that GPS is the GNSS that provides the lowest and most stable TTFF for all attenuation levels up to 20 dB. GLONASS behaves in a similar way to GPS until 8 dB, but its performance degrades for higher attenuation. On the other hand, BeiDou and Galileo present TTFF performances that are significantly affected by attenuation.

### 3.4. Positioning Error

Obtaining a low TTFF is useful as long as the accuracy of the position provided by the receiver is sufficient for the intended purposes of the corresponding application. For this reason, to evaluate the robustness of the TTFF, the positioning error obtained in the first fix also needs to be analysed. Taking into account that most mass-market applications are performed on the ground, this study specifically focuses on the horizontal error. The horizontal error is derived by converting the first fixed position, calculated through standard point positioning (SPP), into Universal Transverse Mercator (UTM) coordinates and comparing them with the georeferenced position of the GNSS antenna.

In Figure 8, the outcome of the horizontal error as a function of attenuation is depicted for the different GNSSs. The associated statistics (mean, standard deviation, minimum, and maximum) are also provided in Table 3. In those cases where no fix is achieved, the horizontal error assigned is the highest value obtained in any other run of the same recording for that specific GNSS. However, this limitation imposed on the maximum value gives rise to an artificial reduction in the standard deviation for high attenuation levels.

Under optimal conditions, all GNSSs achieve similar results regarding the horizontal error, except for GLONASS. GLONASS presents larger errors, which are already above 5 metres without any attenuation. As attenuation increases, GPS exhibits the lowest degradation of its horizontal positioning. BeiDou and Galileo also present acceptable results, with those of Galileo being more stable. However, their horizontal errors increase with attenuation at a higher rate in comparison with those of GPS.

## 4. Discussion

Based on the test results provided above, the robustness of the TTFF for the different GNSSs is discussed here. This test reproduces typical conditions of mass-market applications, relying on the main band of all GNSSs, i.e., B1, L1/E1, and G1, and a low-cost receiver. Thus, the absolute values obtained can only be used as reference for those cases with similar conditions. For this reason, since all four GNSSs have undergone the exact same test conditions, the aim here is to perform a comparative analysis between them.

### 4.1. BeiDou

The number of tracked satellites provided by BeiDou under optimal conditions, which is similar to that of GLONASS, decreases rapidly with attenuation. In this sense, a relevant factor is the reduced visibility, in terms of both time and elevation angle, of the IGSO satellites from the test location.BeiDou is the GNSS that provides the highest signal power in optimal conditions, although it is overcome by GPS and equalized by Galileo when attenuation reaches 8 dB. The same reason commented in the previous point could also be the origin of this behaviour.The TTFF results of BeiDou are acceptable with no attenuation, but they deteriorate when any attenuation is applied, especially for values above 8 dB. Therefore, there seems to be some correlation between this degradation of the TTFF and the already shown decrease in the SNR with attenuation. On the other hand, its horizontal error is not significantly affected by attenuation until about 14 dB.

### 4.2. Galileo

Galileo is the GNSS that exhibits the lowest number of tracked satellites by far, mainly due to the fact it has not yet reached its full operational capability and therefore the number of operational satellites in orbit is lower compared to the other GNSSs.The signal power offered by Galileo in optimal conditions is lower than the one provided by BeiDou, similar to that of GLONASS and higher than that of the GPS one. As long as attenuation increases, its SNR values outperform those of GLONASS and equalize with BeiDou, but are surpassed by the signal power of GPS.The TTFF outcome of Galileo is the worst of all GNSSs, affected by the low availability of satellites. In this sense, Galileo implemented in August 2023 an improvement in its navigation message (I/NAV) that reduces the TTFF [35], but it is common that low-cost receivers are not updated with this new capability yet. However, regarding positioning accuracy, in those cases where a fix is achieved, the horizontal error obtaine is solid, providing values that are more precise than those of BeiDou, but larger than the GPS ones.

### 4.3. GLONASS

The number of tracked satellites of GLONASS is similar to that provided by BeiDou and only behind the performance of GPS. Moreover, the quantity of tracked satellites is less affected by attenuation compared to BeiDou and Galileo.GLONASS is the second best GNSS in terms of SNR under optimal conditions, being overcome only by BeiDou. However, its signal power is the one most affected by attenuation compared to the other three GNSSs.GLONASS provides the lowest TTFF of all GNSSs until 4 dB of attenuation, but above that level, only GPS achieves better results. However, this fast TTFF is obtained at the expense of a high horizontal error, which also degrades remarkably when attenuation is applied.

### 4.4. GPS

GPS provides the largest quantity of tracked satellites of all GNSSs, even though it is not the GNSS with the most operational satellites in orbit. Therefore, although BeiDou provides a similar number of signals (despite those from its IGSO satellites arriving with lower power), the receiver acquires more from GPS satellites. The reason for this could be a higher robustness of GPS signals or a better adaptation of the receiver to them due to GPS legacy.The average signal power of GPS is the smallest under optimal conditions, but this is due to the fact that a higher number of satellites are tracked, even those with low SNR values. In fact, when attenuation increases, its signal power resists better than those of the other GNSSs, surpassing the values of GLONASS and Galileo at 4 dB, and those of BeiDou at 8 dB.GPS presents the lowest TTFF results for attenuation levels above 6 dB. In addition, GPS provides the most accurate horizontal position for all attenuation values.

## 5. Conclusions

In this work, a comparative analysis of the TTFF robustness of the four available GNSSs (BeiDou, Galileo, GLONASS, and GPS), based on their performance against different levels of signal attenuation, up to 20 dB, is presented. This study is limited to a representative low-cost receiver, specifically the u-blox ZED-F9P, and the main GNSS band, as these are typical conditions of many mass-market applications. Due to these constraints, the results are not discussed in absolute terms but as a comparative assessment of the four GNSSs. Nonetheless, since the test proposed here relies on the well-known GSV and GGA NMEA messages, it can be easily reproduced with other receivers and bands.

The results indicate that GLONASS achieves the fastest TTFF when attenuation is low enough, but it also presents the largest horizontal error. On the other hand, BeiDou and Galileo exhibit higher TTFF levels, but constrain better the positioning error. Furthermore, all these three show a significant increase in their TTFF for high attenuation values. Finally, GPS provides the best trade-off between TTFF and accuracy for all attenuation levels. Thus, it can be concluded that GPS is nowadays the GNSS that offers the most robust TTFF for low-cost single-frequency receivers.

## Figures and Tables

**Figure 1 sensors-25-01599-f001:**
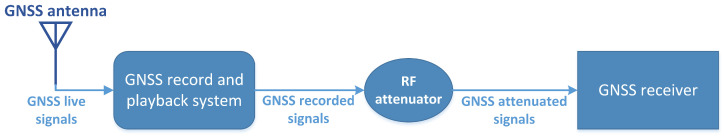
Diagram of the test setup.

**Figure 2 sensors-25-01599-f002:**
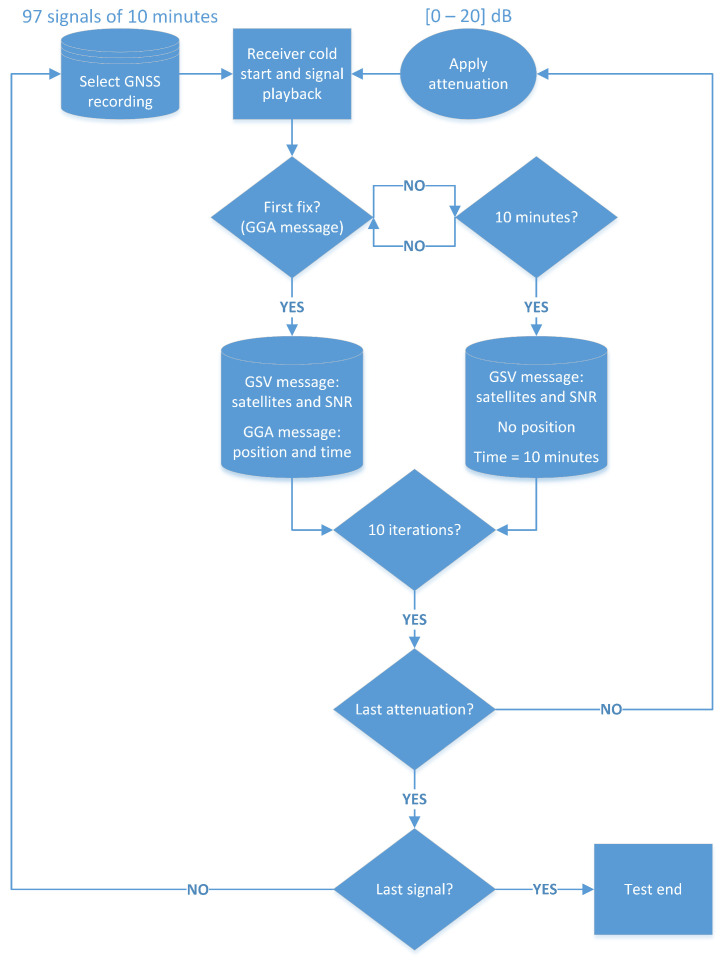
Test methodology workflow for each GNSS.

**Figure 3 sensors-25-01599-f003:**
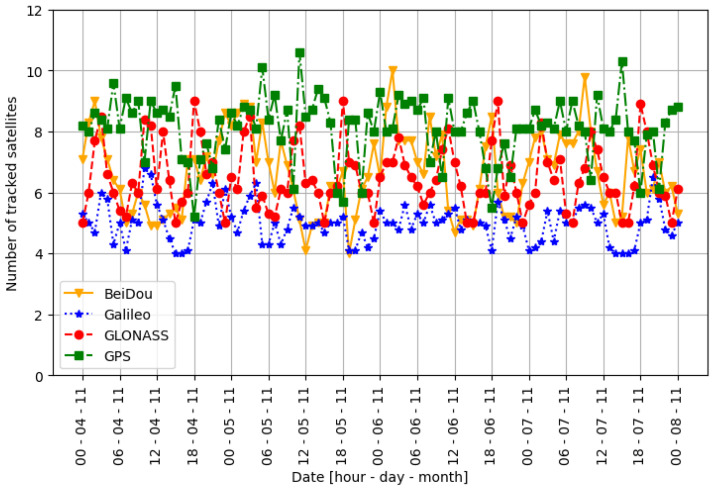
Satellites tracked along the 4 days of the test for the different GNSSs without attenuation.

**Figure 4 sensors-25-01599-f004:**
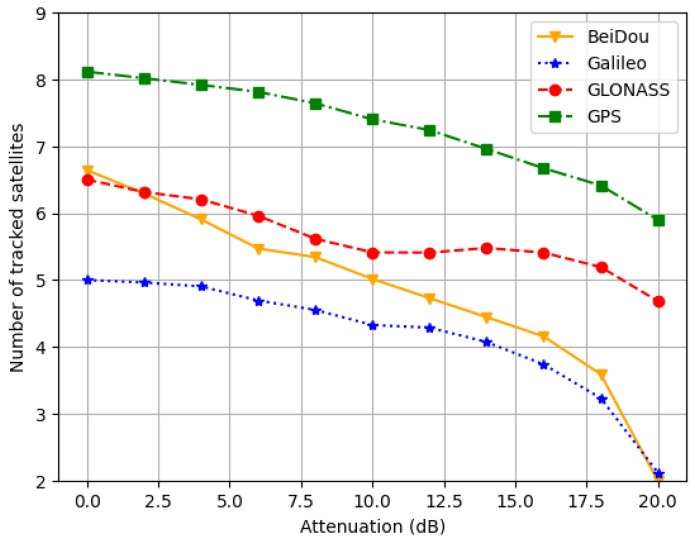
Tracked satellites versus attenuation for the different GNSSs.

**Figure 5 sensors-25-01599-f005:**
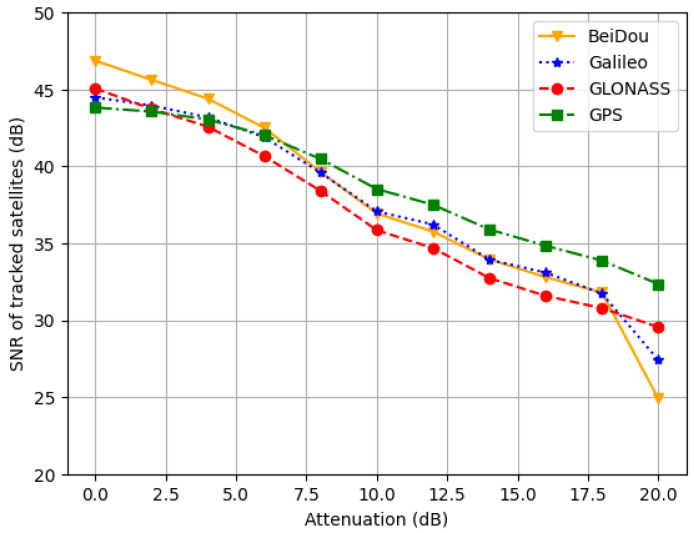
SNR versus attenuation for the different GNSSs.

**Figure 6 sensors-25-01599-f006:**
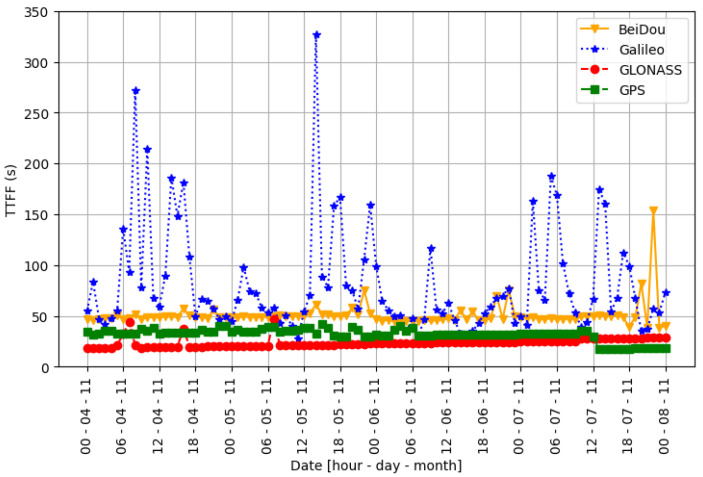
TTFF along the 4 days of the test for the different GNSSs without attenuation.

**Figure 7 sensors-25-01599-f007:**
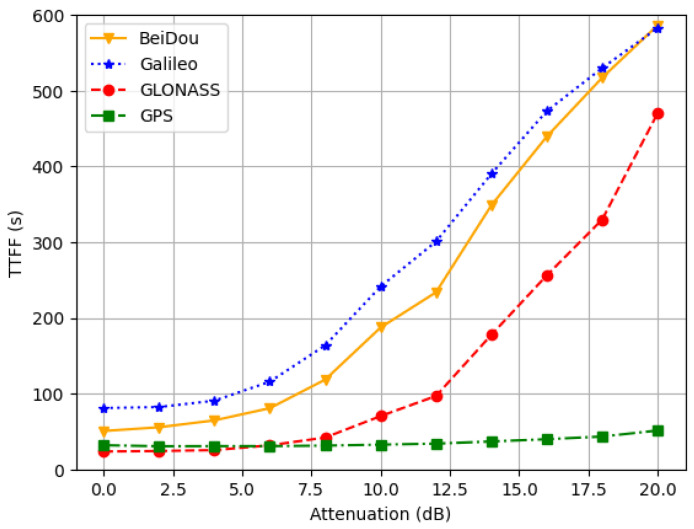
TTFF versus attenuation for the different GNSSs.

**Figure 8 sensors-25-01599-f008:**
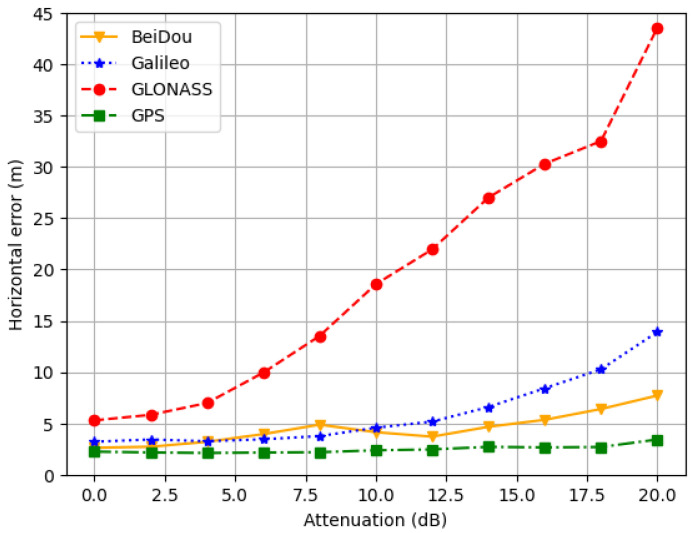
Horizontal error versus attenuation for the different GNSSs.

**Table 1 sensors-25-01599-t001:** Statistical results of tracked satellites without attenuation.

GNSS	Tracked Satellites			
	Mean	Standard Deviation	Minimum	Maximum
BeiDou	6.6	1.3	4.0	10.0
Galileo	5.0	0.6	4.0	6.8
GLONASS	6.5	1.1	5.0	9.0
GPS	8.1	1.0	5.2	10.6

**Table 2 sensors-25-01599-t002:** Statistical results for the TTFF.

TTFF	GNSS	Attenuation (dB)										
		0	2	4	6	8	10	12	14	16	18	20
Mean (s)	BeiDou	50.6	55.6	64.7	80.7	118	187	234	349	439	517	586
	Galileo	80.9	82.2	90.7	116	164	241	301	390	473	530	583
	GLONASS	23.6	24.2	25.4	31.7	42.0	70.3	96.9	178	256	330	470
	GPS	31.9	30.5	30.6	30.7	31.3	32.6	33.9	36.8	39.8	43.3	51.3
	BeiDou	12.4	26.5	49.8	66.1	70.3	88.1	100	127	128	95.8	27.8
Standard	Galileo	52.9	50.5	58.2	77.0	108	138	149	140	107	88.7	39.2
deviation (s)	GLONASS	4.62	5.30	7.79	28.7	65.3	96.4	126	182	202	203	175
	GPS	5.92	4.97	4.78	4.78	4.20	2.82	2.39	3.06	4.11	6.08	12.5
Minimum (s)	BeiDou	38.5	41.8	45.1	48.0	53.6	76.3	117	167	198	273	447
	Galileo	28.2	30.5	32.2	32.4	42.4	73.0	83.1	113	247	273	404
	GLONASS	18.1	18.1	18.1	18.1	18.6	20.1	20.1	25.8	30.5	43.9	63.0
	GPS	17.4	17.4	17.4	17.4	18.1	21.9	27.4	32.1	33.0	34.9	37.4
Maximum (s)	BeiDou	154	225	384	589	600	600	600	600	600	600	600
	Galileo	327	327	380	422	547	600	600	600	600	600	600
	GLONASS	46.3	49.5	72.7	210	600	600	600	600	600	600	600
	GPS	42.0	35.2	35.3	35.2	40.4	40.9	43.5	46.5	60.0	70.8	119

**Table 3 sensors-25-01599-t003:** Statistical results for the horizontal error.

Horizontal Error	GNSS	Attenuation (dB)										
		0	2	4	6	8	10	12	14	16	18	20
Mean (m)	BeiDou	2.62	2.72	3.20	3.94	4.86	4.12	3.89	5.61	9.34	12.2	14.8
	Galileo	3.21	3.43	3.26	3.66	4.17	5.16	6.12	7.74	9.97	11.6	13.3
	GLONASS	5.29	5.83	6.98	9.94	13.5	18.6	22.0	27.0	30.3	32.5	43.6
	GPS	2.24	2.16	2.11	2.15	2.19	2.37	2.46	2.71	2.66	2.69	3.41
Standard deviation (m)	BeiDou	1.62	1.71	2.96	5.18	7.51	3.23	2.60	5.38	17.9	19.4	20.0
	Galileo	3.65	4.12	3.79	4.65	5.33	7.10	8.23	9.39	11.3	11.5	11.7
	GLONASS	3.94	4.70	6.14	10.4	11.0	15.7	20.7	27.3	30.8	30.9	37.1
	GPS	1.47	1.37	1.32	1.35	1.36	1.60	1.78	1.79	1.71	1.83	2.30
Minimum (m)	BeiDou	0.22	0.37	0.44	0.50	0.42	0.67	0.70	0.54	0.74	0.94	2.03
	Galileo	0.25	0.32	0.46	0.63	0.62	0.71	0.64	0.52	1.22	1.52	1.52
	GLONASS	0.62	0.71	1.10	1.56	1.68	1.78	4.28	4.04	2.59	3.69	3.49
	GPS	0.21	0.21	0.22	0.26	0.41	0.23	0.40	0.85	0.59	0.58	0.99
Maximum (m)	BeiDou	11.5	10.7	26.3	48.6	66.4	21.2	13.4	32.3	163	163	163
	Galileo	29.3	31.0	26.5	27.4	36.6	43.1	43.1	54.6	59.2	59.2	59.2
	GLONASS	30.5	37.5	46.8	63.3	69.2	72.2	114	155	173	155	226
	GPS	8.30	6.71	6.45	7.26	7.59	8.45	10.5	10.3	9.80	12.6	17.0

## Data Availability

In order to access the dataset of recorded signals, please contact the corresponding author directly. Due to the large size of the files, it is not feasible to store them in a public repository. Arrangements for data transfer will be agreed upon request.

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
