# Peer review of "Time to First Fix Robustness of Global Navigation Satellite Systems: Comparison Study"

_sensors, 2025, doi:10.3390/s25051599_

Round 1

Reviewer 1 Report

Comments and Suggestions for Authors

The study analyzes the the time to first fix (TTFF) of BeiDou, Galileo, GLONASS and GPS, using a low-cost receiver. The paper can be published after a minor revision, addressing a few simple questions raised below.

L. 33. This implies that, for a receiver, it may be easier to acquire and track signals to calculate its first position when using some GNSS in comparison to others, mainly under adverse conditions. This aspect is especially important for the low-cost GNSS chipsets of mass-market devices, which usually undergo economic, size or power constraints.

-- This immediately induces the question, whether the results of the study should depend on a specific brand of the receiver.

L. 101. Moreover, in order to detect as many satellites as possible, no minimum values for the elevation angle and the signal to noise ratio (SNR) are set in its navigation input filters. On the other hand, all the parameters of its navigation output filters are kept with their default values, so the quality of the solution is not affected.

-- Not affected by what? Low elevation angles and low SNR should affect it. Right?

L. 138. Afterwards, the performance of the TTFF itself is assessed.

-- What does it mean? TTFF itself is a measure of the receiver performance.

-- Figure 5: Some explanations are needed. One would expect a very simple linear relation between attenuation and SNR: SNR(Attenuation) = SNR(0) - Attenuation. The dependencies shown are not equal to this and different for different GNSSs. Why?

-- Figure 3: Why not provide a single panel with all the four days? What do non-integer number mean? This also applies to Table 1. I understand that the mean and STD do not have to be integer, but what about minimum and maximum?

Author Response

Comments 1: L. 33. This implies that, for a receiver, it may be easier to acquire and track signals to calculate its first position when using some GNSS in comparison to others, mainly under adverse conditions. This aspect is especially important for the low-cost GNSS chipsets of mass-market devices, which usually undergo economic, size or power constraints.

-- This immediately induces the question, whether the results of the study should depend on a specific brand of the receiver.

Response 1: Thank you for the comment, as it is indeed an important point. The u-blox ZED-F9P receiver is widely used in the literature for studies addressed to low-cost GNSS devices. Regarding this matter, we included in the paper some references from Sensors itself (Janos et al. 2021, Robustelli et al. 2023, Tomastik et al. 2023), but there exist many more from other scientific journals. In addition, as it is stated in the paper (page 2, lines 72-79), the objective is to perform a comparative assessment between GNSSs (not to discuss their absolute figures), which we consider that is achieved by testing the signals of all four GNSSs under the very same conditions.

Comments 2: L. 101. Moreover, in order to detect as many satellites as possible, no minimum values for the elevation angle and the signal to noise ratio (SNR) are set in its navigation input filters. On the other hand, all the parameters of its navigation output filters are kept with their default values, so the quality of the solution is not affected.

-- Not affected by what? Low elevation angles and low SNR should affect it. Right?

Response 2: Thank you for pointing this out, as we now realize that, without further explanation, that statement may be confusing. The navigation output filters of the receiver are not related to elevation angles and SNR. These filters control the quality of the PVT, so if the solution does not comply with the specified conditions, it is not provided to the user. Actually, the only filters of interest for this test are the maximum value of the dilution of precision and the minimum (estimated) positioning accuracy. Therefore, we added this information in the new version of the paper to make it clearer: page 3, lines 115-117.

Comments 3: L. 138. Afterwards, the performance of the TTFF itself is assessed.

-- What does it mean? TTFF itself is a measure of the receiver performance.

Response 3: We wanted to say that in the previous sections we have analysed factors that affect the TTFF, such as the number of tracked satellites and the signal power, and now we will study the TTFF itself that is achieved with the different GNSSs. Perhaps the use of the word “performance” is misleading. In this sense, we have modified that sentence in the new versión of the paper: page 5, lines 150-151.

Comments 4: -- Figure 5: Some explanations are needed. One would expect a very simple linear relation between attenuation and SNR: SNR(Attenuation) = SNR(0) - Attenuation. The dependencies shown are not equal to this and different for different GNSSs. Why?

Response 4: The attenuation is not strictly linear because it depends on the set of satellites that the receiver tracks each time, which is not always the very same. Considering your equation, it means that we do not have in every case the same SNR(0). Some reasons for that are the following. There is some randomness involved in the tracking and acquisition algorithms of the receiver (whose influence we have tried to reduce performing 10 runs per case). For instance, for cold start, it may not start searching satellites in the same order. This effect is more relevant with low attenuattion, as the receiver can see more satellites with enough SNR and it is not obliged to track the same set for the first fix. In addition, for higher attenuations, satellites whose SNR was close to the limit eventually cannot be acquired anymore and are replaced after some time by new more powerful ones. These same reasons influence the differences between GNSSs. To try to clarify this matter, we have included a brief justification in the new version of the paper: page 7, lines 106-108

Comments 5: -- Figure 3: Why not provide a single panel with all the four days?

Response 5: Thank you for the comment. The reason is that we had issues with the generation of the ticks and grid of the figure when using the data of all four days and it was not so visually appealing. However, thanks to your sugestion, we have adressed again the problem and finally achieved a good solution. Therefore, we have modified Figure 3, following your proposal, in the new version of the paper: page 6.

Comments 6: What do non-integer number mean? This also applies to Table 1. I understand that the mean and STD do not have to be integer, but what about minimum and maximum?

Response 6: The results are the average of the 10 runs per scenario (a 10-minute scenario every hour). As a different number of satellites can be tracked in each run (due to what was commented in response 4), the results can be non-integer. Therefore, the minimum and maximum are the worst and best scenarios of these averages of 10 runs. On the other hand, the mean and std correspond to these averages of 10 runs from all the 97 scenarios.

Reviewer 2 Report

Comments and Suggestions for Authors
  1. It is suggested that the author can describe the requirements on TTFF of low-cost receivers of market segments mentioned in this paper. The impact of TTFF on these applications.
  2.  It is suggested that the author not only consider the position errors but also the timings errors of different systems. 
  3. Please confirm whether the  u-blox ZED-F9P receiver used in this paper can simultaneously receive BDS-2 and BDS-3 signals in this paper. 
  4. This paper consider the signal deterioration impact on TTFF, it is suggested that authors further explain  the relationship between the SNR value and the degree of deterioration of the electromagnetic environment. 
  5. As this paper only consider one type of low-cost receiver,  it should highlight in the conclusion part that the conclusions is valid only for this type of receiver. 

Author Response

Comments 1: It is suggested that the author can describe the requirements on TTFF of low-cost receivers of market segments mentioned in this paper. The impact of TTFF on these applications.

Response 1: Thank you for the suggestion. We agree with it, so in the new version of the paper we have included a description of the advantages of having a low TTFF for mass-market applications and their impact on the segments mentioned: page 2, lines 46-58

Comments 2: It is suggested that the author not only consider the position errors but also the timings errors of different systems.

Response 2: We have considered your suggestion and assessed the possibility of studying the time error with acquired the test data. However, we have realised that due to the lack of an accurate time reference in the employed setup, we cannot obtain reliable results related to timing. Anyway, we highly appreciate the suggestion, as it is a interesting point that we will take into account for our future works.    

Comments 3: Please confirm whether the  u-blox ZED-F9P receiver used in this paper can simultaneously receive BDS-2 and BDS-3 signals in this paper. 

Response 3: Yes, we confirm that the u-blox ZED-F9P receiver acquires and tracks simultaneously operational satellites of all generations, not only for BeiDou but for the other GNSSs as well. In this sense, a explicit clarification has been added in the new version of the paper: page 3, lines 110-111.

Comments 4: This paper consider the signal deterioration impact on TTFF, it is suggested that authors further explain  the relationship between the SNR value and the degree of deterioration of the electromagnetic environment. 

Response 4: Thank you for the suggestion. We agree with it, so in the new version of the paper we have included an explanation of the environmental causes of SNR deterioration: page 6, lines 183-189.

Comments 5: As this paper only consider one type of low-cost receiver,  it should highlight in the conclusion part that the conclusions is valid only for this type of receiver. 

Response 5: Thank you for the suggestions. We agree with it and the text of the conclusion has been modified to clarify this matter: page 13, lines 314-316. 

Reviewer 3 Report

Comments and Suggestions for Authors

This paper presents a study comparing the robustness of Time to First Fix (TTFF) across different GNSS constellations. However, there are several aspects that need clarification:

  1. The TTFF of a constellation (BeiDou, Galileo, GLONASS, and GPS) depends on the location of the receiver, the position of the satellites, and the survey environment. This means that comparisons should take into account the receiver's location, the time of day (which affects the satellite sky map), and the PDOP (Position Dilution of Precision). With varying locations and times, the results may differ.

  2. What is the GNSS record-and-playback system, and how does it work? (Line 105)

  3. Should this manuscript be considered a case report?

  4. In Figure 5, is the SNR of the tracked satellites the average of all satellites?

  5. Regarding Figure 6, is the attenuation the average of all the satellites? Also, could you provide an additional figure showing TTFF versus time?

Comments on the Quality of English Language

 English could be improved to more clearly express the research.

Author Response

Comments 1: The TTFF of a constellation (BeiDou, Galileo, GLONASS, and GPS) depends on the location of the receiver, the position of the satellites, and the survey environment. This means that comparisons should take into account the receiver's location, the time of day (which affects the satellite sky map), and the PDOP (Position Dilution of Precision). With varying locations and times, the results may differ.

Response 1: Thank you for your comment. You are completely right, for this reason, in order to take into account those factors, the test covers 4 complete days, so many different geometries of the satellites are included and 11 levels of attenuation, so different kinds of hostile environments are simulated (see pages 3-4: lines 118-130). In addtion, the study follows a comparative approach based on average terms and it does not discusses the TTFF absolute performance for each GNSSs, as we understand that the obtained values depend also on the specifc technical equipment (see page 2, lines 76-79)   

Comments 2: What is the GNSS record-and-playback system, and how does it work? (Line 105)

Response 2: We agree with your comment, so, for clarification, in the new version of the paper we have provided the specific model of the system and included a reference to its manual: page 3, lines 89-91  

Comments 3: Should this manuscript be considered a case report?

Response 3: We do not consider it as a case report, because we have approached the study as a general assessment, in comparative terms, of all GNSSs, not as a specific case. For this reason, we have covered as many scenarios as possible, including 4 complete days of signals and a wide attenuation range [0-20 dB]. In addition, although only one receiver is employed, we selected the u-blox ZED-F9P because it is widely used in the literature for studies addressed to low-cost GNSS devices. Regarding this matter, we included in the paper some references from Sensors itself (Janos et al. 2021, Robustelli et al. 2023, Tomastik et al. 2023), but there exist many more from other scientific journals.

Comments 4: In Figure 5, is the SNR of the tracked satellites the average of all satellites?

Response 4: Yes, as it is stated in the text (page 7, lines 194-196): “The SNR values are obtained as the mean of the SNR levels of all the tracked satellites, provided by the GSV NMEA message, at the time of the first fix or at the end of the test run (10 minutes) if no fix is achieved”. However, if we are missing something and some further explanation is needed, please let us know.

Comments 5: Regarding Figure 6, is the attenuation the average of all the satellites? Also, could you provide an additional figure showing TTFF versus time?

Response 5: Regarding the attenuation, not only of Figure 6 but of all other Figures, it is applied with a RF attenuator to the recorded signals (see Figure 1), so it affects equally to all satellites. Regarding the suggestion of the additional figure, we agree with it and we have included it in the new version of the paper as new Figure 6.

Comments on the Quality of English Language: English could be improved to more clearly express the research.

Response on the Quality of English Language: We have performed a complete language review for the new version of the paper and implemented some corrections. Anyway, we are more than open to assess and improve any specific language issue that can be pointed out.

Round 2

Reviewer 3 Report

Comments and Suggestions for Authors

The authors' response addressed all of my concerns, and I accept this version of this manuscript.